# Neural Network Based Model Predictive Control for a Quadrotor UAV

**Bailun Jiang** [1], **Boyang Li** [1,2,*], **Weifeng Zhou** [3], **Li-Yu Lo** [1], **Chih-Keng Chen** [4] **and Chih-Yung Wen** [1,2]

1. Department of Aeronautical and Aviation Engineering, The Hong Kong Polytechnic University, Kowloon, Hong Kong
2. Research Centre for Unmanned Autonomous Systems, The Hong Kong Polytechnic University, Kowloon, Hong Kong
3. School of Professional Education and Executive Development, The Hong Kong Polytechnic University, Kowloon, Hong Kong
4. Department of Vehicle Engineering, National Taipei University of Technology, Taipei 10608, Taiwan
* Correspondence: bo-yang.li@polyu.edu.hk

**Abstract:** A dynamic model that considers both linear and complex nonlinear effects extensively benefits the model-based controller development. However, predicting a detailed aerodynamic model with good accuracy for unmanned aerial vehicles (UAVs) is challenging due to their irregular shape and low Reynolds number behavior. This work proposes an approach to model the full translational dynamics of a quadrotor UAV by a feedforward neural network, which is adopted as the prediction model in a model predictive controller (MPC) for precise position control. The raw flight data are collected by tracking various pre-designed trajectories with PX4 autopilot. The neural network model is trained to predict the linear accelerations from the flight log. The neural network-based model predictive controller is then implemented with the automatic control and dynamic optimization toolkit (ACADO) to achieve real-time online optimization. Software in the loop (SITL) simulation and indoor flight experiments are conducted to verify the controller performance. The results indicate that the proposed controller leads to a 40% reduction in the average trajectory tracking error compared to the traditional PID controller.

**Keywords:** feedforward neural network; model predictive control; UAV; trajectory tracking; position control

## 1. Introduction

Recently, unmanned aerial vehicles (UAVs), especially multi-rotors, are developing fast due to the advanced microcontroller and sensing technologies. Small UAVs are widely used for infrastructure inspection, land surveying, and search and rescue [1,2]. Precise trajectory tracking is a crucial requirement for UAVs operating under various conditions to ensure a safe and efficient mission. Versatile methods have been proposed to improve the position tracking accuracy of quadrotor UAV subject to input delay, model uncertainties, and wind disturbances [3–5]. Many researchers proposed using the model predictive control (MPC) for UAVs to achieve precise trajectory tracking performance [6–8]. MPC is a feedback optimal control method that takes the system model into account based on a receding horizon principle. The general working principle of MPC is using a system model to predict the future behavior of a process and calculate the optimal control inputs under actuator constraints. The optimizer obtains an optimal control sequence, and only the first value is applied to the system. This online optimization process is conducted repeatedly at each timestep. MPC has been widely used in the process industry due to its ability to handle multi-input multi-output (MIMO) systems with constraints [9–11]. Another advantage of MPC is the preview feature. The traditional PID controller corrects the states based on tracking error, which inevitably results in a tracking delay. MPC is able to consider the reference trajectory and predicted system response in the prediction horizon.

Model-based control algorithms, such as MPC, extensively benefit from an accurate system model. A mathematical model which describes the dominant mechanical dynamics of a quadrotor is established based on the Newton–Euler formalism. However, measuring the model parameters ,such as the moment of inertia and propeller lift coefficient through experiments, could be challenging. In this condition, system transfer functions and state-space models can be identified using input and output data. This process, known as system identification, is widely used for UAVs because of its simplicity and effectiveness [8,12,13]. The limitation of this approach is that the identified result is confined by the state-space model and transfer function, which make it hard to model the complicated nonlinear aerodynamic effects. Hoffmann et al. [14] and Fay [15] studied the aerodynamic effects of a quadrotor, such as hub force, rolling moment effect, and blade flapping. These effects introduce extra forces and moments to quadrotor dynamics but are usually too complicated to be identified by both linear and nonlinear models.

Lately, learning-based methods have been studied to model quadrotor dynamics and improve controller performance. Bansal et al. [16] modeled quadrotor dynamics by a feed-forward neural network (FFNN) known as rectified-linear unit (ReLU) and then employed the network in the LQR controller. Torrente et al. [17] used Gaussian processes to complement the nominal dynamics of the quadrotor in an MPC pipeline. Bauersfeld et al. [18] model the quadrotor with blade-element-momentum theory and compensate the residual dynamics with temporal-convolutional (TCN) encoders. These works show that learning-based methods have good potential to model the complicated aerodynamic effects of the quadrotor. Because the neural network is developed as the MPC prediction model, the complexity is confined by controller sample time and platform computational capability. A deep RNN is capable of learning explicit system behaviors, such as motor delay and complex aerodynamics. However, such a network is hard to be deployed for real-time calculation or used in optimal control problems (OCP). To achieve a trade-off between model accuracy and simplicity, FFNN is adopted instead of RNN in the current study.

By using learning-based methods, the traditional UAV modeling processes such as wind tunnel experiments, computational fluid dynamics (CFD) simulation, and derivation of dynamic equations can be extensively simplified. In this article, a neural network-based MPC (NNMPC) is developed for quadrotor position tracking. Without prior knowledge of quadrotor dynamic equations, the proposed method models full translational dynamics of the quadrotor purely from flight data. The NNMPC is implemented with the automatic control and dynamic optimization (ACADO) toolbox for real-time online computation. The tracking performance is evaluated with new trajectories different from the training samples. Both simulation and experiment are conducted and the results from different controllers are compared. The main contributions of this article include:

- Learning the full translational dynamics of a quadrotor purely from flight data without prior knowledge of quadrotor dynamic equations. The proposed model balances the trade-off between model accuracy and simplicity.
- Synthesizing the proposed FFNN with the MPC scheme for the real-time position control of a quadrotor.
- Demonstrating the validity and control performance in simulation and real-world flight experiments by comparison with the PID controller and nonlinear MPC (NMPC).

The rest of this article is organized as follows. In Section 2, the governing equations of quadrotor dynamics are reviewed and a neural network modeling method is proposed. In Section 3, NNMPC is formulated based on the network prediction model. In Section 4 and Section 5, the controller performance is tested in simulation and flight experiments, respectively, followed by the conclusion in Section 6.

## 2. Neural Network Model of Quadrotor UAV

This section introduces the general quadrotor dynamic equations, which describe the dominant forces and moments experienced by the UAV during flight and help us to choose eligible inputs of the network model. Then, we present a neural network architecture to

model the quadrotor dynamics. A shallow feedforward network scheme is adopted due to its accurate prediction and simple structure.

### 2.1. Quadrotor Dynamic Model

The quadrotor UAV is usually assumed to be a rigid body with six degrees of freedom. The system state vector is defined as $\mathbf{X} = [x\ y\ z\ u\ v\ w\ \phi\ \theta\ \psi\ p\ q\ r]^T$, where $x$, $y$, $z$ and $u$, $v$, $w$ denotes position and velocity in the North-East-Down inertial frame $\Gamma_I$. $\phi$, $\theta$, $\psi$ denotes Euler angles in roll, pitch, and yaw axes, respectively, and $p$, $q$, $r$ denotes angular velocities in the body frame $\Gamma_B$, respectively. Figure 1 shows the quadrotor coordinate systems.

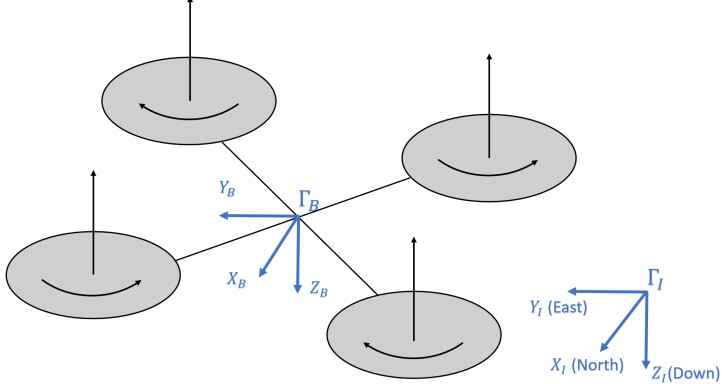

**Figure 1.** The quadrotor sketch with inertial frame $\Gamma_I$ and body frame $\Gamma_B$.

The system is controlled by four control inputs $\mathbf{U} = [U_1, U_2, U_3, U_4]$, where $U_1$ is the thrust force along the $Z_B$ direction, and $U_2$, $U_3$, $U_4$ are rolling, pitching, and yawing moments in $\Gamma_B$, respectively. Quadrotor nonlinear dynamic model is constructed based on the Newton–Euler formalism [19]:

$$
\begin{cases}
\dot{x} = u \\
\dot{y} = v \\
\dot{z} = w \\
\dot{u} = -(\cos\phi\sin\theta\cos\psi + \sin\phi\sin\psi)\dfrac{U_1}{m} \\
\dot{v} = -(\cos\phi\sin\theta\cos\psi - \sin\phi\cos\psi)\dfrac{U_1}{m} \\
\dot{w} = -(\cos\phi\cos\theta)\dfrac{U_1}{m} + g \\
\dot{\phi} = p + (\sin\phi\tan\theta)q + (\cos\phi\tan\theta)r \\
\dot{\theta} = q\cos\phi - r\sin\phi \\
\dot{\psi} = \dfrac{\sin\phi}{\cos\theta}q + \dfrac{\cos\phi}{\cos\theta}r \\
\dot{p} = \dfrac{(I_y - I_z)qr + U_2}{I_x} \\
\dot{q} = \dfrac{(I_z - I_x)pr + U_3}{I_y} \\
\dot{r} = \dfrac{(I_x - I_y)pq + U_4}{I_z}
\end{cases}
\tag{1}
$$

where $m$ is the mass of the quadrotor, $g$ is the gravitational acceleration, and $I_X$, $I_Y$, $I_Z$ are quadrotor moment of inertia around three axes.

The nonlinear model in (1) is widely used as the prediction model in the MPC scheme. Note that this model only takes gravitational force and control inputs into consideration and neglects influences from other sources, such as air drag, the gyroscope effect, and the

hub force. Because the MPC performance relies on a high accuracy prediction model, we aim to develop a neural network model to identify full translational dynamics of the quadrotor UAV, including all the aerodynamic effects.

### 2.2. Neural Network Structure

In recent years, the spectrum of machine learning has significantly advanced thanks to the development of computing technologies. As a subfield of machine learning, supervised learning is proven to be effective for tasks such as object detection and natural language processing. The key feature of supervised learning is that the desired output is available during the training process. After learning from the examples, the network is expected to approximate the system behavior and predict system outputs by the input data that it has not been trained with. Motivated by the universal approximation feature of the neural network, we trained an FFNN with the supervised learning approach to predict system states in the future.

In FFNN, there is no recurrent feedback loop. Therefore, a well-trained FFNN has fixed weights and biases, representing a static mapping from inputs to outputs. Apart from FFNN, RNN is another potential candidate for modeling dynamic systems. The output signal of RNN is connected back to the input ports with a tapped delay line (TDL). This implies that the output of RNN depends both on its input and previous output. The recursive structure enables RNN to learn complex system dynamics with input delay. The architectures of FFNN and RNN are presented in Figure 2. We chose FFNN instead of RNN for two main reasons. First, the prediction model in MPC is written in state-space form, which predicts the state derivative of the next step based on the current state and control input. FFNN is capable of predicting such a system with good accuracy. Furthermore, the training procedure of RNN is much more complex and time consuming than that of FFNN because issues such as gradient explosion and state initialization greatly influence training results [20,21].

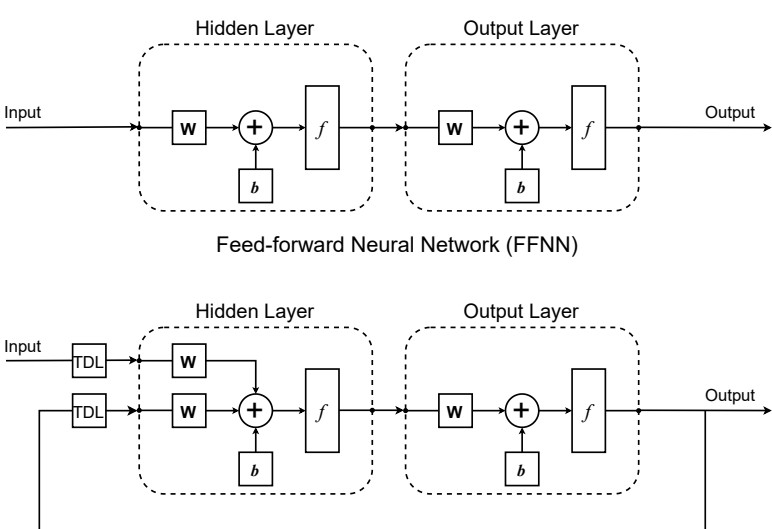

**Figure 2.** Comparison of neural network architecture between FFNN and RNN.

Our network model is a two-layer FFNN with one hidden layer and one output layer, where sigmoid and pure linear transfer functions are used, respectively. The network model can be expressed as:

$$\alpha = w^T \xi \left( W^T \beta + B \right) + b, \tag{2}$$

where $\beta$ represents network input, which consists of states and control inputs. The weight matrices are denoted by $W$ and $w$ whereas the bias vectors are denoted by $B$ and $b$, where

uppercase is for the hidden layer and lowercase is for the output layer. $\xi$ is the sigmoid activation function in the hidden layer. The pure linear activation function is adopted in the output layer.

Then, we chose the input and output data for the network. Because this work focuses on developing the outer loop position controller for a quadrotor, only translational dynamics is required to be identified. In other words, the translational acceleration of three axes will be modeled by the network using current states as inputs. We used thrust, Euler angles, and linear velocities as inputs to the network. Based on (1), the acceleration $\dot{u}$, $\dot{v}$, and $\dot{w}$ are mainly dependent on thrust and Euler angles. Taking velocities as input enables the network to identify the air drag model as drag is considered to be proportional to velocity [8]. To simplify the network model, position and angular velocity are not used as input because they are not directly related to translational dynamics. The final network model has seven inputs and three outputs, which can be written as:

$$[\dot{u}_{NN}\ \dot{v}_{NN}\ \dot{w}_{NN}] = f_{NN}(u, v, w, \phi, \theta, \psi, U_1) \tag{3}$$

where $\dot{u}_{NN}$, $\dot{v}_{NN}$, and $\dot{w}_{NN}$ represent predicted accelerations and $f_{NN}$ represents the neural network transfer function.

The training objective is to determine the weights and biases that minimize the mean squared error (MSE) between predicted acceleration $\hat{h} = [\dot{u}_{NN}\ \dot{v}_{NN}\ \dot{w}_{NN}]$ and observed acceleration $h = [\dot{u}\ \dot{v}\ \dot{w}]$ subject to (2):

$$min \sum_{k=1}^{N} \frac{1}{N} \left\| \hat{h} - h \right\|^2. \tag{4}$$

Once the training process is complete, the weights and biases will be fixed, and the network can be deployed to predict system dynamics from new inputs. The details of the neural network training process will be discussed in Sections 4 and 5.

## 3. Model Predictive Controller Design

Two main components in MPC are the prediction model and the online optimizer. The system behavior under certain control inputs is calculated by the prediction model, which in our work is an FFNN. The optimizer solves the quadratic programming (QP) problem, formulated as:

$$\begin{aligned}
min \quad & \int_{t=0}^{T} ||h(x(t), u(t)) - y_{ref}||_Q^2 dt \\
& + ||h(x(T)) - y_{N,ref}||_{Q_N}^2 dt \\
s.t. \quad & \dot{x} = f(x(t), u(t)) \\
& u(t) \in \mathcal{U} \\
& x(t) \in \mathcal{X} \\
& x(0) = x(t_0),
\end{aligned} \tag{5}$$

where $u(t)$ and $x(t)$ denotes control input and state at timestep $t$, $T$ denotes the number of timesteps that the model predicts, which is also known as prediction horizon, $y_{ref}$ and $y_{N,ref}$ denotes reference state for prediction horizon and terminal timestep, $Q$ and $Q_N$ denotes weighting matrices for states and terminal states, $f(\cdot)$ and $h(\cdot)$ denote the prediction function and system output function, $\mathcal{U}$ and $\mathcal{X}$ represents input constraint and state constraint, respectively.

The OCP in (5) is solved by the multiple shooting method. The system is discretized from $t_0$ to $t_T$ and a boundary value problem (BVP) is formulated at each time interval with variable constraints imposed. The BVP is solved with the sequential quadratic programming (SQP) technique by the active set method using the qpOASES solver [8,22]. MPC is a receding horizon control technique which means that only the first value of the optimized

control sequence is applied to the system whereas the rest of it is regarded as an initial guess for the OCP in the next iteration.

The quadrotor controller is usually implemented in the cascaded loop scheme. The outer loop controller tracks position reference with thrust and Euler angle commands whereas the inner loop controller tracks attitude reference with moments in the corresponding axis. This work focuses on the development of the outer loop position controller, which is an MPC using FFNN as the prediction model. A standard PID controller is adopted for inner loop attitude control. The cascaded control structure adopted in this work is shown in Figure 3.

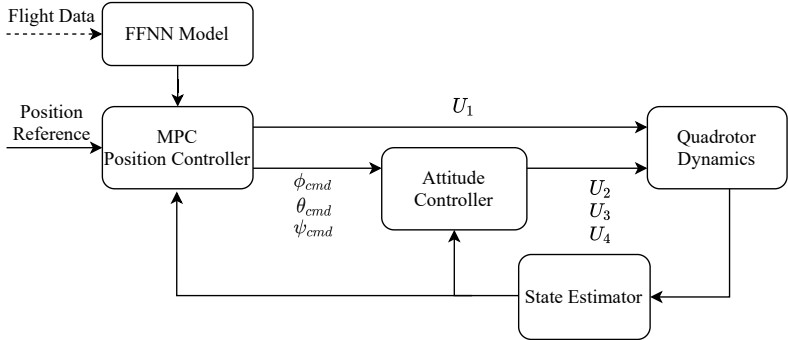

**Figure 3.** Cascaded loop control structure for quadrotor UAV.

The prediction model of the position MPC controller is formulated as:

$$
\begin{cases}
\dot{x} = u \\
\dot{y} = v \\
\dot{z} = w \\
\dot{u} = NN_X \\
\dot{v} = NN_Y \\
\dot{w} = NN_Z \\
\dot{\phi} = \dfrac{\phi_{cmd} - \phi}{\tau_\phi} \\
\dot{\theta} = \dfrac{\theta_{cmd} - \theta}{\tau_\theta}
\end{cases}
\tag{6}
$$

where $NN_X$, $NN_Y$, and $NN_Z$ are linear accelerations in the respective axes predicted by the neural network. Each acceleration prediction will be integrated to obtain the corresponding velocity and integrated again to the position. Because the error accumulates through the integration process, an accurate prediction of acceleration is required for position tracking.

Note that prediction model in (6) linearizes the inner loop dynamics by first-order transfer functions, where $\tau_\phi$ and $\tau_\theta$ are time constants of roll and pitch control, $\phi_{cmd}$ and $\theta_{cmd}$ are roll and pitch commands sent to inner loop attitude control. The values of $\tau_\phi$ and $\tau_\theta$ can be derived with flight data by the system identification technique. The yaw angle is assumed to be zero during the entire flight so that the yaw angle is excluded from MPC states.

## 4. Simulation Results

To validate the algorithm, we carry out software in the loop (SITL) simulations by the PX4 open-source flight control platform [23], using an Iris quadrotor model in the Gazebo robot simulator shown in Figure 4. The simulated quadrotor communicates with PX4 using MAVLink API, which defines a set of messages to supply sensor data from the simulated world to PX4 and return motor and actuator values applied to the simulated vehicle [24]. For a qualitative evaluation of NNMPC developed in this work, we simulated the same tracking flight using PID, NMPC, and NNMPC controllers.

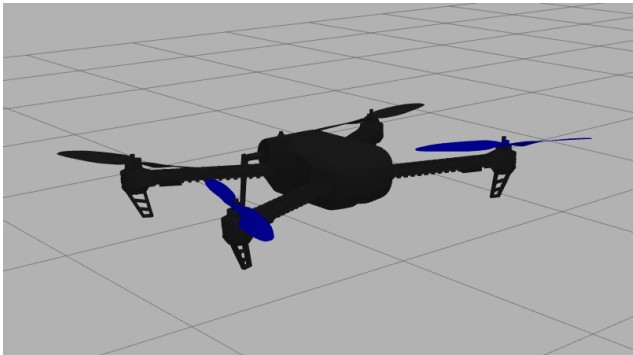

**Figure 4.** Iris quadrotor UAV model in Gazebo simulator.

*4.1. Data Collection*

A good training dataset should contain enough training samples and cover as many operating conditions of the quadrotor as possible. Therefore, several training trajectories, such as random sinusoidal waves, a step reference, and a circular path, were used to generate training data. The designed trajectory was tracked by the PID controller provided by PX4 autopilot. Note that the yaw angle command $\psi_{cmd}$ was set to be zero throughout the flight so that the $x$ and $y$ position references were tracked by pitch and roll movements, respectively. High precision tracking is not required during this process because we aim to collect the raw control input and state output samples, which are not related to the outer loop control scheme. The quadrotor states were logged at 100 Hz and synchronized at 10 Hz before network training. The training flight log contains 600 s of flight data, corresponding to 6000 data samples.

*4.2. Neural Network Training*

An FFNN was trained to learn the linear acceleration in all three axes with the collected flight data by minimizing the MSE between target and predicted accelerations. We used current velocities, Euler angles, and thrust as input and current linear accelerations as target outputs. The flight data were filtered and normalized prior to network training. A low pass filter was applied to the data to remove high-frequency noises, which are usually observed in thrust data. The training inputs and targets were scaled so that each channel has a zero mean and a unity standard deviation. This guarantees that the network equally weights all inputs and targets during the training process. Seventy percent of the collected data was used for training, 15% was used for validation, and the remaining 15% was used for testing.

We invoke the MATLAB Deep Learning Toolbox [25] to train the network by the Levenberg–Marquardt backpropagation algorithm. The minimum gradient, number of hidden layers, number of neurons in hidden layers, and maximum epochs were set at 0, 1, 10, and 1000, respectively. The training process of such a shallow neural network with few neurons took several minutes, depending on the size of the training data. The network was trained with random initial weights and the best normalized MSE obtained was 0.00491. Once the training was complete, the network model could be expressed by substituting weights and bias in (2). Figure 5 compares the neural network model prediction outputs with the measured accelerations on the test data which were not used for training. The result indicates that the neural network model is able to predict current accelerations with given inputs to good accuracy. However, the results in $X_I$ and $Y_I$ directions are generally better than those in the $Z_I$ direction. The reason is that the collected thrust data contains high-frequency noise and data integrity could be violated during the filtering process. Because thrust is the dominant input for $Z_I$ acceleration prediction, the result in $Z_I$ direction is affected most by inaccurate thrust training data. For $X_I$ and $Y_I$ directions, on the other hand, the dominant inputs are the Euler angles so that the results are not affected as much. We further found out that $X_I$ and $Y_I$ accelerations can be accurately predicted only using velocities and Euler angles as inputs.

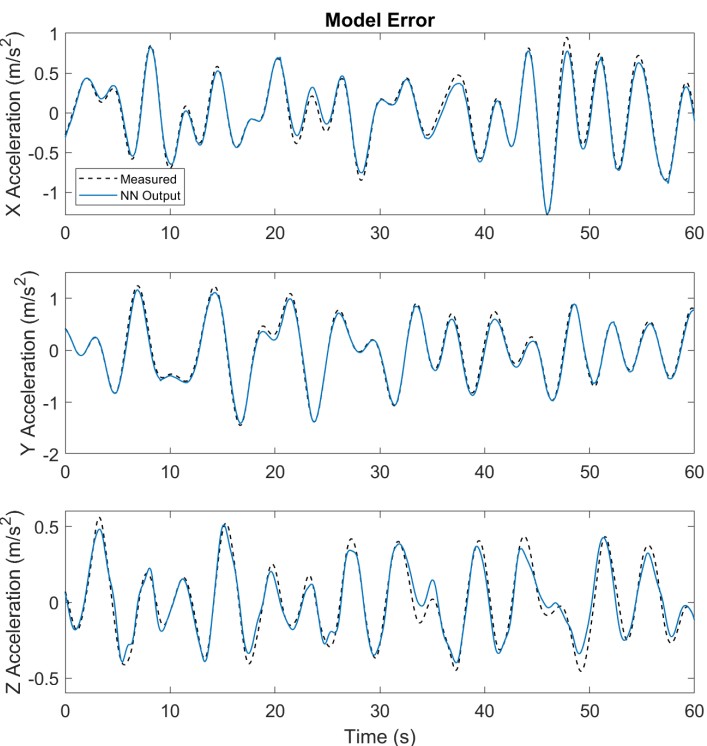

**Figure 5.** Comparison of neural network model prediction output and measured data for linear accelerations from Gazebo Iris quadrotor UAV.

*4.3. MPC Implement*

We next developed NMPC and NNMPC for quadrotor position tracking. The nonlinear prediction model in NMPC is formulated as:

$$
\begin{cases}
\dot{x} = u \\
\dot{y} = v \\
\dot{z} = w \\
\dot{u} = -(\cos\phi\sin\theta\cos\psi + \sin\phi\sin\psi)\dfrac{U_1}{m} \\
\dot{v} = -(\cos\phi\sin\theta\cos\psi - \sin\phi\cos\psi)\dfrac{U_1}{m} \\
\dot{w} = -(\cos\phi\cos\theta)\dfrac{U_1}{m} + g \\
\dot{\phi} = \dfrac{\phi_{cmd} - \phi}{\tau_\phi} \\
\dot{\theta} = \dfrac{\theta_{cmd} - \theta}{\tau_\theta}
\end{cases}
\tag{7}
$$

where $\tau_\phi$ and $\tau_\theta$ share the same values as in (6). The MPC runs at 20 Hz with 1 s ($20 \times 0.05$ s) prediction horizon. The roll angle and pitch angle were constrained in the range of $[-20, 20]$ degrees. Both NMPC and NNMPC controllers used the same parameters, so the tracking results were only influenced by the different acceleration prediction models.

MPC solves OCP at each timestep and usually involves extra computational load. To achieve real-time calculation, both NMPC and NNMPC were implemented using the ACADO toolbox to generate a fast C-code solver [26]. ACADO has a MATLAB interface that exports the solver as a MEX file, which is combined with the MAVLink subscriber and publisher in the Simulink environment. The neural network model function was generated by MATLAB *genFunction* syntax and used in NNMPC as the prediction model.

### 4.4. Trajectory Tracking Results

Next, we conducted the SITL simulation to compare the tracking performance of different control algorithms. Two trajectories were designed to compare the control performance of PID, NMPC, and NNMPC controllers. The first trajectory consists of a two-meter step reference in both $X_I$ and $Y_I$ directions and a one-meter step reference in the $Z_I$ direction. The second trajectory consists of sinusoidal waves with increasing frequency in the range of $[0.5, 0.85]$ Hz. The yaw angle reference was set to be zero throughout the flight and the inner attitude control loop was handled by a PID controller.

The step trajectory tracking results of PID, NMPC, and NNMPC are compared in Figure 6. Generally speaking, MPCs are superior to the PID controller because of less tracking delay. The preview feature of MPCs enables them to respond to references in advance. The MPCs have a prediction horizon of 1 s so that they actuate the quadrotor about 0.5 s before the step reference and thus result in an early step response with less tracking error.

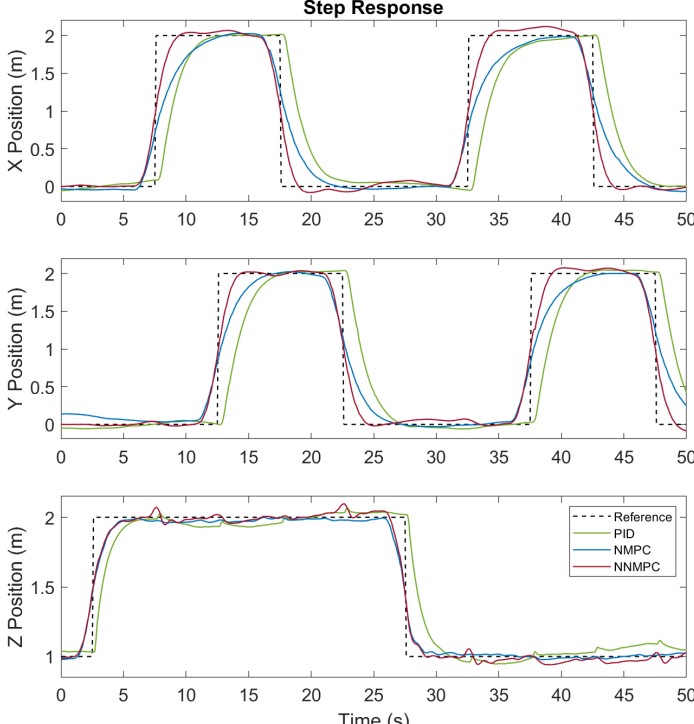

**Figure 6.** Step trajectory Tracking results for PID, NMPC, and NNMPC controllers in three axes from simulation.

By comparing the results of NMPC and NNMPC, we found out that the NNMPC has faster transient response in $X_I$ and $Y_I$ directions. Because NMPC and NNMPC are using the same parameters, the improved performance results from the more accurate neural network prediction model. However, in the $Z_I$ direction, there is no obvious difference between NMPC and NNMPC results. It is probably because the nonlinear prediction model already describes the system dynamics to a quite accurate extent. Furthermore, a minor coupling between lateral and vertical movement is noticed in the $Z_I$ direction for NNMPC. Because we predicted accelerations in all axes with one neural network, such coupling cannot be completely eliminated but is attenuated to an acceptable degree. Table 1 summarizes the Root Mean Square Error (RMSE) for all controllers tracking results. For step trajectory, NNMPC improves the RMSE of NMPC by 30% and 25% in $X_I$ and $Y_I$ directions, respectively.

Figure 7 depicts the sinusoidal trajectory tracking result of PID, NMPC, and NNMPC. The results indicate that the PID controller has around 1 s of tracking delay whereas that of MPCs is almost invisible. The sinusoidal trajectory presents a more obvious comparison

between NMPC and NNMPC. Figure 8 shows the trajectory tracking results of NMPC and NNMPC plotted in 3D view. Because the PID results in 2 s of tracking delay, its trajectory is not plotted here. With increasing velocity, the trajectory of NMPC deviates from the reference path and results in a circular trajectory with a smaller radius compared with the NNMPC trajectory. According to the tracking result RMSE in Table 1, NNMPC reduces the RMSE of NMPC by 49% and 65% in $X_I$ and $Y_I$ directions, respectively. The simulation results thus validated that the neural network-based model predicted system dynamics well and improved the MPC performance.

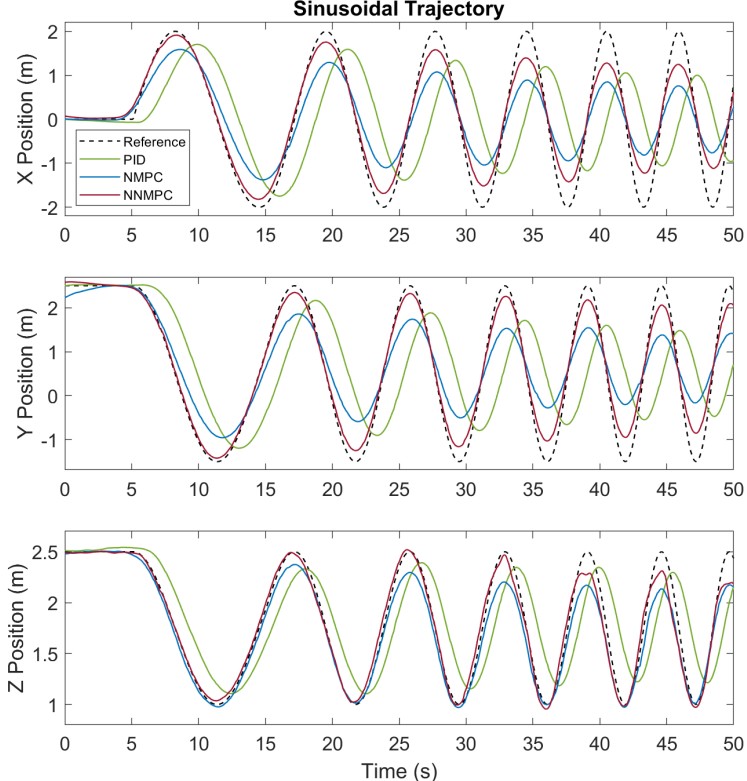

**Figure 7.** Sinusoidal trajectory tracking result for PID, NMPC, and NNMPC controllers in three axes from simulation.

**Table 1.** RMSE of Trajectory Tracking from Simulation Result.

| Trajectory Type | Direction | PID (m) | NMPC (m) | NNMPC (m) |
|---|---|---|---|---|
| Step | X | 0.597 | 0.344 | 0.240 |
| | Y | 0.595 | 0.326 | 0.245 |
| | Z | 0.164 | 0.087 | 0.086 |
| Sinusoidal | X | 1.340 | 0.634 | 0.324 |
| | Y | 1.354 | 0.642 | 0.225 |
| | Z | 0.347 | 0.138 | 0.092 |

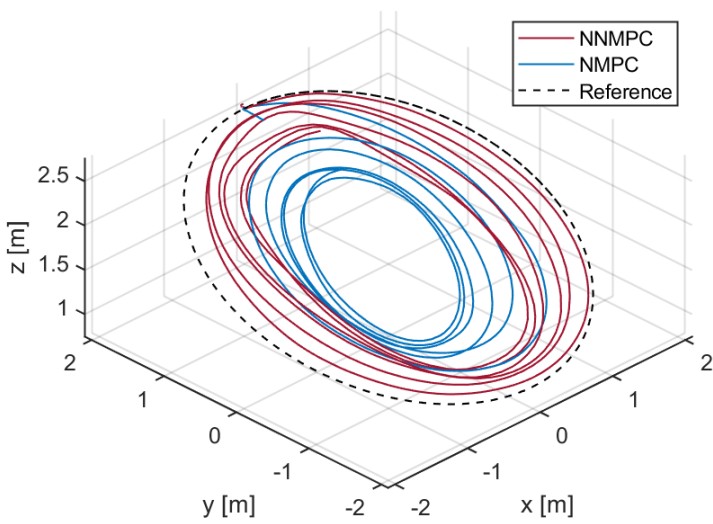

**Figure 8.** 3D view of the sinusoidal trajectory tracking result for NMPC and NNMPC in simulation.

## 5. Flight Experiments

### 5.1. Neural Network Modeling

After the improved performance of NNMPC was validated by simulation, we further collected the training data with an F330 quadrotor UAV shown in Figure 9. The training trajectories were identical to those used in the previous simulations. The Robot Operating System (ROS) framework was adopted for communication between the quadrotor and the ground station.

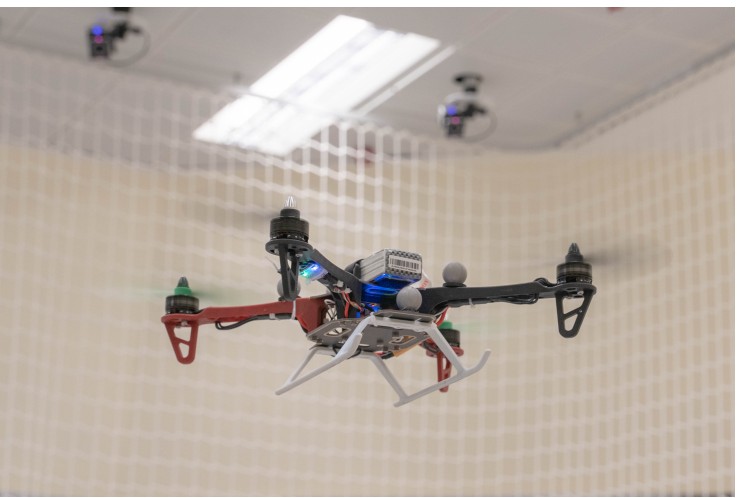

**Figure 9.** F330 quadrotor UAV used for flight experiments.

UAV states were logged by IMU running at 250 Hz and an external motion capture system VICON running at 100 Hz. The recorded flight data from different sensors were synchronized at 10 Hz. The duration of training flight data was 600 s, which corresponds to 6000 input and target data samples for the neural network.

The network was trained by the same method as Section IV.B. Figure 10 shows the prediction result of the trained network with an MSE of 0.0279. We discovered that compared to the Gazebo simulation, the real-world flight log contains more noise, which results in the training MSE being about five times higher than simulation training results.

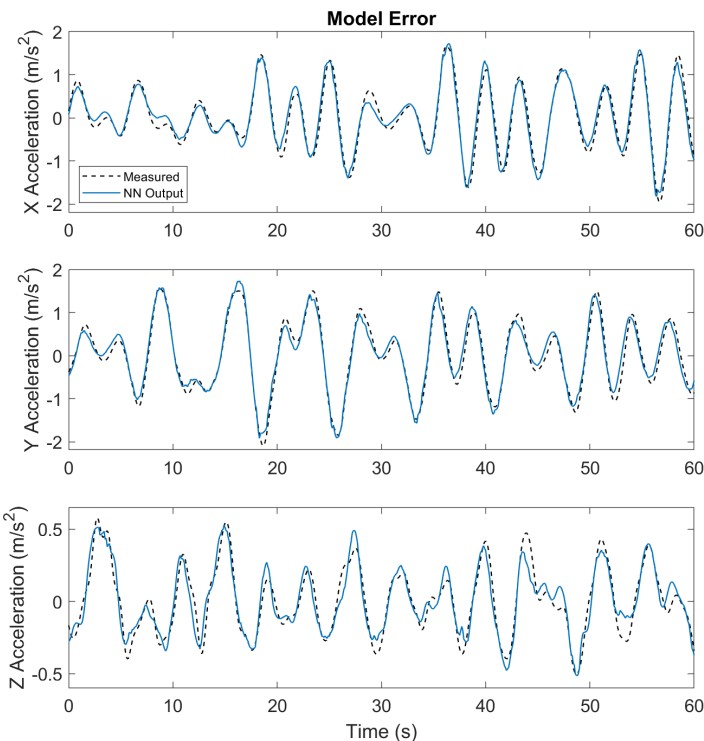

**Figure 10.** Comparison of neural network model prediction output and measured data for linear accelerations from the F330 quadrotor UAV.

In the MPC optimization process, the predicted accelerations are integrated through the prediction horizon, which indicates that the error of predicted acceleration could lead to a significant position error given a period. Because the logs from flight experiments include more noise from sensors, the training network was modified to eight hidden neurons to avoid overfitting and maintain good prediction results.

The network was also trained by different learning algorithms, training parameters, or datasets, such as *cos* and *sin* values. However, as the network has a relatively simple structure, the training scheme contributes little to the prediction results. Collecting high-quality training data is the crucial step and a long flight log that covers more operating conditions with a lower noise level will extensively improve NNMPC controller performance.

### 5.2. Trajectory Tracking Results

The NMPC and NNMPC controllers were implemented on a ThinkPad X1 Carbon laptop computer (Intel i5-7200U @ 2.50GHz) for flight experiments. Because ACADO transformed the control algorithm in C-code, the computational load is relatively low. For the 20 Hz update frequency and $20 \times 0.05$ s prediction horizon, both MPCs result in solver time less than 5 ms and the maximum CPU load recorded during the flight is 30%. The details of the computation complexity of the proposed NNMPC are summarized in Table 2. Note that the OCP time is the average solving time recorded during the experiment. The reference values for velocity are calculated by the time derivative of position reference and the Euler angle references are set to be zero to achieve a stable flight.We test the control performance with the same testing trajectory as in simulation. A video showing the experiment results is available at: https://youtu.be/KYH02a_53fs (accessed on 19 August 2022).

**Table 2.** Proposed NNMPC Parameters and OCP Performance.

| | |
|---|---|
| Prediction horizon | 20 |
| Sample time (s) | 0.05 |
| $Q$ | [12 12 4 1 1 1 1 1 200 10 10] |
| $Q_N$ | [12 12 4 1 1 1 1 1] |
| OCP time (ms) | 4 |

The trajectory tracking results of PID, NMPC, and NNMPC for the step response are shown in Figure 11. Similar to the simulation results, PID control results have apparent position tracking delays whereas two MPC methods eliminate the delay with the preview feature. In addition, NNMPC outperforms NMPC and PID with the shortest delay and rise time due to its accurate neural network-based prediction model. According to the RMES results summarized in Table 3, NNMPC improved 15% and 12% in $X_I$ and $Y_I$ directions compared with NMPC results.

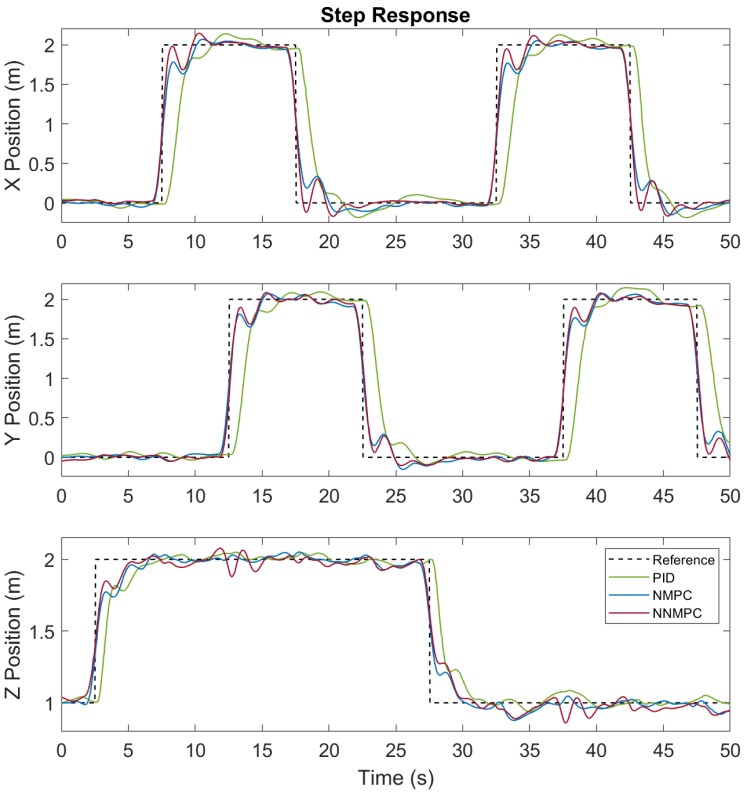

**Figure 11.** Step trajectory tracking results for PID, NMPC, and NNMPC controllers in three axes from the flight experiment.

The sinusoidal trajectory tracking results of PID, NMPC, and NNMPC are compared in Figure 12. In general, the controller performances in the flight experiment are quite similar to the result from the simulation. However, in the flight experiment, the difference between NMPC and NNMPC is attenuated. NNMPC still outperforms NMPC in $X_I$ and $Y_I$ directions but in the $Z_I$ direction the performance is almost identical. In the experiment, the neural network model accuracy is compromised by the more noisy training data and thus the NNMPC performance is slightly influenced. Figure 13 shows the trajectory tracking results of NMPC and NNMPC plotted in 3D view. Because the PID results in 2 s of tracking delay, its trajectory is not plotted here. The figure shows that at relatively low velocity, both NMPC and NNMPC track the reference with good accuracy. However, with increasing velocity, the nonlinear model fails to predict the system states precisely and leads to larger

tracking error of NMPC, whereas the NNMPC trajectory is closer to the reference. Table 3 shows that NNMPC reduces the RMSE of NMPC by 47% and 39% in $X_I$ and $Y_I$ directions.

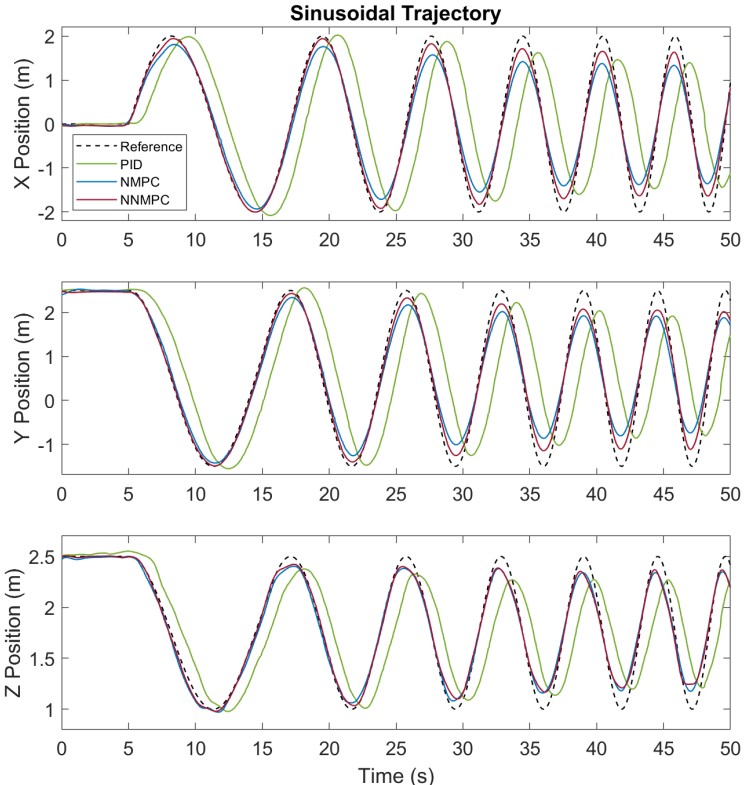

**Figure 12.** Sinusoidal trajectory tracking result for PID, NMPC, and NNMPC controllers in three axes from real-world experiment.

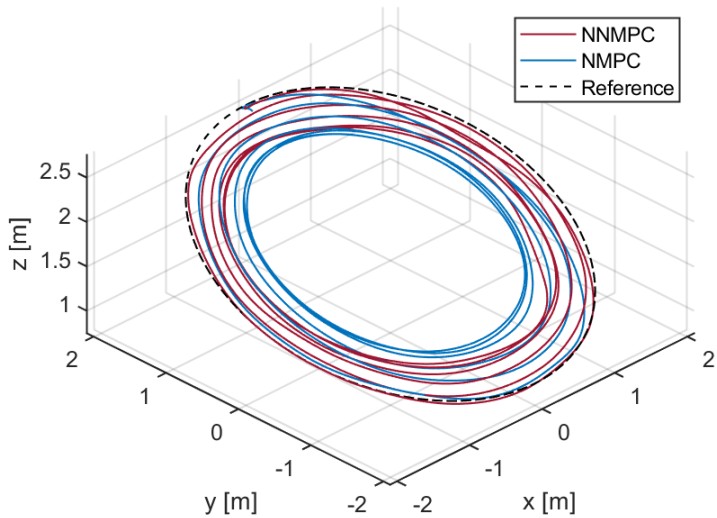

**Figure 13.** 3D view of the sinusoidal trajectory tracking result for NMPC and NNMPC in flight experiment.

**Table 3.** RMSE of Trajectory Tracking from Simulation Result.

| Trajectory Type | Direction | PID (m) | NMPC (m) | NNMPC (m) |
|:---:|:---:|:---:|:---:|:---:|
| | X | 0.525 | 0.208 | 0.176 |
| Step | Y | 0.542 | 0.204 | 0.193 |
| | Z | 0.173 | 0.092 | 0.010 |
| | X | 1.177 | 0.312 | 0.164 |
| Sinusoidal | Y | 1.179 | 0.320 | 0.196 |
| | Z | 0.341 | 0.088 | 0.087 |

Both the Gazebo simulation and flight experiment results demonstrate improved control performance for the NNMPC controller, indicating that:

- FFNN is able to predict the UAV dynamics beyond the training data. The network learned a dynamic model from training data and extrapolated it with good accuracy.
- The shallow network structure with one hidden layer and around eight neurons is simple enough to be implemented in MPC for real-time OCP calculation and yet accurate enough to predict the system dynamics.
- MPC benefits from a neural network prediction model trained by flight logs. By adopting a network model instead of a nonlinear model, the average tracking error is attenuated by around 40%.

## 6. Conclusions

In this work, FFNN was used to predict full translational dynamics and integrated with the MPC framework for a quadrotor UAV. Based on the obtained flight data, FFNN learns the dynamic equations and enhances the MPC performance. The simulation and experiment results show that the neural network-based model predictive controller improves position tracking performance. Furthermore, the proposed approach is fully based on flight data and no mathematical dynamic equations are involved. For non-conventional UAV configurations with substantial aerodynamic contribution, such as tail-sitter and tilt-rotor vertical takeoff and landing UAVs, the proposed neural network-based MPC would be more applicable than traditional methods because the nonlinear aerodynamic terms at different flight conditions (hovering, level flight, and transition phase) for those UAVs are hard to build.

Future work could include modeling the system with different types of neural networks to improve the prediction performance and training the prediction model online to adapt to varying conditions, such as a battery voltage drop or different wind disturbances. In addition, the proposed method could be applied to other UAV configurations with more complicated aerodynamic effects.

**Author Contributions:** Conceptualization, B.J., L.-Y.L. and W.Z.; methodology, B.J., L.-Y.L. and W.Z.; software, B.J. and W.Z.; validation, B.J. and L.-Y.L.; formal analysis, B.J. and B.L.; investigation, B.J. and B.L.; resources, B.J. and B.L.; data curation, B.J.; writing—original draft preparation, B.J.; writing—review and editing, C.-Y.W., C.-K.C. and B.L.; visualization, B.J.; supervision, C.-Y.W., C.-K.C. and B.L.; project administration, C.-Y.W., C.-K.C. and B.L.; funding acquisition, C.-Y.W. and B.L. All authors have read and agreed to the published version of the manuscript.

**Funding:** This research was funded by the PolyU Start-up Fund (P0034164, P0036092) and Research Centre for Unmanned Autonomous Systems (P0042699).

**Institutional Review Board Statement:** Not applicable.

**Informed Consent Statement:** Not applicable.

**Data Availability Statement:** Not applicable.

**Conflicts of Interest:** The authors declare no conflict of interest.

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
