# Peer review of "Neural Network Based Model Predictive Control for a Quadrotor UAV"

_aerospace, doi:10.3390/aerospace9080460_

Round 1

Reviewer 1 Report

I think that this paper should be revised carefully before a decision about the publication of this paper. The suggested comments are as follows:

-         The authors should polish the paper suitably. The whole paper should be reviewed carefully, in order to correct all the typing errors.

-         In introduction, it is not enough to state the current work. It should be expended and reconstructed. Including the motivation, the main difficulties, the main work and the improvements compared with previous related works should be emphasized in this section.

-         There are no numerical simulation results, which can demonstrate the superiority of the proposed control law. I suggest comparing the simulations with the results of the recent (2021-2022) related valid references.

-         The novelty of the proposed method should be highlighted carefully.

-         More figures should be added to further demonstrate the effectiveness of the proposed method.

-         The main background of this paper is not clear. The authors should highlight the main background of this paper. The background information is very useful for the understanding of this paper. Some new advances in this field are missing, such as: Mathematics 10 (10), 1659; ISA transactions 123, 455-471;        Aerospace Science and Technology 121, 107337     

-         The importance of the problem considered in this paper should be further addressed. 

-         The directions to further and improve the work should be added as future recommendation section after ‘conclusions’ section.

-         No issue regarding complexity of the proposed method has been presented.

-         A movie of experimental results in YouTube can be very good for the understanding of the success of controller.

According to my comments to the paper, I recommend revision of this paper.

Reviewer 2 Report

Authors proposed neural network based model predictive control for a quadrotor UAV. An approach was proposed to model full translational dynamics of a quadrotor UAV by a feedforward neural network, which is adopted as the prediction model in a model predictive controller (MPC) for precise position control. The results indicate that the proposed controller leads to a 40 % reduction in the average trajectory tracking error compared to the traditional PID controller. However, some suggestions are,

(1)    The introduction is too long to focus on the research problems.

(2)    In Figure 6, the responses by NMPC and NNMPC are ahead of the reference signal. Why? .

(3)     In Z Position of Figure 6, the system errers of NNMPC is larger than PID. Why?

Round 2

Reviewer 1 Report

The revised paper is appropriate now. No more comments.